# Exosomes in the Diagnosis and Treatment of Renal Cell Cancer

**DOI:** 10.3390/ijms241814356

**Published:** 2023-09-20

**Authors:** Stergios Boussios, Perry Devo, Iain C. A. Goodall, Konstantinos Sirlantzis, Aruni Ghose, Sayali D. Shinde, Vasileios Papadopoulos, Elisabet Sanchez, Elie Rassy, Saak V. Ovsepian

**Affiliations:** 1Department of Medical Oncology, Medway NHS Foundation Trust, Gillingham ME7 5NY, UK; aruni.ghose1@gmail.com (A.G.); elisabet.sanchez@nhs.net (E.S.); 2Faculty of Life Sciences & Medicine, School of Cancer & Pharmaceutical Sciences, King’s College London, Strand, London WC2R 2LS, UK; 3Kent Medway Medical School, University of Kent, Canterbury CT2 7LX, UK; 4AELIA Organization, 9th Km Thessaloniki–Thermi, 57001 Thessaloniki, Greece; 5School of Sciences, Faculty of Engineering and Science, University of Greenwich, Chatham Maritime ME4 4TB, UK; p.devo2@greenwich.ac.uk (P.D.); c.a.i.goodall@greenwich.ac.uk (I.C.A.G.); s.v.ovsepian@greenwich.ac.uk (S.V.O.); 6School of Engineering, Technology and Design, Canterbury Christ Church University, Canterbury CT1 1QU, UK; k.sirlantiz@kent.ac.uk; 7Barts Cancer Centre, Barts Health NHS Trust, London EC1A 7BE, UK; 8Mount Vernon Cancer Centre, East and North Hertfordshire NHS Trust, Northwood HA6 2RN, UK; 9Immuno-Oncology Clinical Network, London, UK; 10Centre for Tumour Biology, Barts Cancer Institute, Cancer Research UK Barts Centre, Queen Mary University of London, London EC1M 6BQ, UK; s.shinde@smd22.qmul.ac.uk; 11Department of Urology, Medway NHS Foundation Trust, Gillingham ME7 5NY, UK; vpapadoster@gmail.com; 12Department of Medical Oncology, Gustave Roussy Institut, 94805 Villejuif, France; elie.rassy@hotmail.com

**Keywords:** renal cell cancer, exosomes, tumor microenvironment, mRNA, miRNA, tumor drug resistance

## Abstract

Renal cell carcinoma (RCC) is the most prevalent type of kidney cancer originating from renal tubular epithelial cells, with clear cell RCC comprising approximately 80% of cases. The primary treatment modalities for RCC are surgery and targeted therapy, albeit with suboptimal efficacies. Despite progress in RCC research, significant challenges persist, including advanced distant metastasis, delayed diagnosis, and drug resistance. Growing evidence suggests that extracellular vesicles (EVs) play a pivotal role in multiple aspects of RCC, including tumorigenesis, metastasis, immune evasion, and drug response. These membrane-bound vesicles are released into the extracellular environment by nearly all cell types and are capable of transferring various bioactive molecules, including RNA, DNA, proteins, and lipids, aiding intercellular communication. The molecular cargo carried by EVs renders them an attractive resource for biomarker identification, while their multifarious role in the RCC offers opportunities for diagnosis and targeted interventions, including EV-based therapies. As the most versatile type of EVs, exosomes have attracted much attention as nanocarriers of biologicals, with multi-range signaling effects. Despite the growing interest in exosomes, there is currently no widely accepted consensus on their subtypes and properties. The emerging heterogeneity of exosomes presents both methodological challenges and exciting opportunities for diagnostic and clinical interventions. This article reviews the characteristics and functions of exosomes, with a particular reference to the recent advances in their application to the diagnosis and treatment of RCC.

## 1. Introduction

Renal cell cancer (RCC) ranks as the third most prevalent cancer affecting the urinary system. According to the World Health Organization, it was the 16th most diagnosed and fatal cancer globally in 2020 [1]. Clear cell RCC, a renal cortical tumor characterized by malignant epithelial cells, is the predominant form of RCC, constituting approximately 80% of all cases [2]. Despite recent advancements, the global incidence and mortality rates of RCC continue to rise, and its prognosis remains unfavorable. The diagnosis and treatment of RCC also entail various societal factors, including racial disparities observed in patients undergoing robotic radical nephrectomy, the primary curative treatment for localized RCC [3,4]. However, approximately 40% of RCC patients experience tumor recurrence following curative surgical resection [5]. Individuals presenting with metastatic RCC or experiencing relapse after local therapy often need systemic treatment. The current repertoire of systemic therapies includes small-molecule tyrosine kinase inhibitors (TKI), cytokines, and monoclonal antibodies, including checkpoint inhibitors, which have been evaluated as both first-line and second-line treatment options [6,7]. Simultaneously, drug resistance in RCC remains a significant factor contributing to treatment failure. Despite the utilization of agents such as vascular endothelial growth factor (VEGF) inhibitors, mammalian target of rapamycin (mTOR) inhibitors, and RAF kinase inhibitors in RCC treatment, the clinical application of these drugs often encounters the challenge of drug resistance [8]. Consequently, early diagnosis and monitoring of RCC can prove advantageous in enhancing clinical management, enabling medical or surgical intervention prior to tumor cell metastasis, and prolonging overall patient survival. Unfortunately, there are currently no specific molecular markers for clinical use in RCC diagnosis [9].

Extracellular vesicles (EVs) present within the tumor microenvironment (TME) are increasingly recognized as significant contributors to various processes, including carcinogenesis, angiogenesis, premetastatic niche (PMN) formation, immune system dysfunction, and dissemination of drug resistance, thereby adding a novel dimension to the complexity of the TME [10]. As a result, the cargo carried by tumor-derived EVs holds great potential as a plentiful source for biomarker discovery. Moreover, the mechanisms involved in the biogenesis, secretion, and uptake of EVs are increasingly understood, offering promising targets for cancer therapy [11]. The term EVs, as defined by the International Society of Extracellular Vesicles (ISEV), encompasses a heterogeneous group of vesicles released by cells that can be categorized into three main types: exosomes, microvesicles, and apoptotic bodies [12]. Exosomes are derived from the inward budding of multivesicular bodies (MVBs) and are small vesicles ranging from ~30 to ~150 nm in diameter. Following the fusion of MVBs with the plasma membrane, exosomes are released into the extracellular space. They are secreted not only by normal human cells but also by tumor cells, and they play a crucial role in facilitating intercellular communication. These small vesicles transport diverse cargo, including proteins, messenger RNAs (mRNAs), microRNAs (miRNAs), and signaling molecules. By carrying and transmitting these signaling molecules, exosomes regulate the physiological state of cells [13].

In 2018, the ISEV published updated guidelines known as Minimal Information for Studies of Extracellular Vesicles (MISEV) with the goal of enhancing the reliability, reproducibility, and acceptance of EV research. These guidelines establish a systematic standard for obtaining EV- and exosome-related data, enabling proper interpretation of research findings [14]. A comprehensive summary of the potential applications of EVs in RCC is currently lacking. Therefore, this review aims to address the knowledge gap by presenting the latest research advancements in the emerging field of EVs in RCC. It provides an overview of the diverse biology, functions, and applications of EVs, with special reference to exosomes in the diagnosis and treatment of RCC.

## 2. Methods

Medline/PubMed and Google Scholar were systematically searched from their inception up to July 2023 for publications in the English language related to exosomes in the diagnosis and treatment of RCC. The search strategy involved primarily the terms (“Exosomes” [Mesh]) AND “Renal Cell Cancer” [Mesh]) in Medline, while Google Scholar was queried with keywords including “Renal Cell Cancer”, “Tumor microenvironment”, “mRNA”, “miRNA”, and “Tumour Drug Resistance”. The screening process involved a manual assessment of article titles and abstracts by SB and SO. Inclusion criteria encompassed peer-reviewed original research publications that featured an appropriate number of subjects. Additionally, reference lists within the retrieved articles were scrutinized to identify any other relevant papers that could augment our review.

## 3. Composition, Secretion, and Detection of Exosomes

Effective intercellular communication is crucial for regulating and coordinating numerous processes within multicellular organisms. Recently, EVs have emerged as an important and highly conserved mechanism for cell signaling and communication in healthy and diseased tissue. Among these vesicles, exosomes, which originate from endosomes—MVBs—and typically have a diameter ranging from 30 to 100 nm, are released into the extracellular environment by cells from diverse tissues and organs [15]. They fulfill multiple roles in different effector mechanisms, serving as mediators of paracrine cell signaling and actively participating in immune regulation [16].

Exosomes possess a unique membrane structure that exhibits resistance to external proteases and RNA enzymes (Figure 1). As a result, they preserve the stability of intracellular functional proteins, mRNAs, and miRNAs. This characteristic renders exosomes highly valuable not only in deciphering the nature of physiological cell–cell signaling but also as sensitive markers for impairments of intercellular communication and diagnosing diseases [17,18]. Exosomes can bring about changes in cellular or tissue states in various medical conditions, making exosome-related assays effective and non-invasive methods for disease diagnosis and monitoring [19]. Furthermore, investigating the molecular mechanisms involved in the exosome-mediated intercellular material exchange will contribute to the theoretical foundation for the development of therapies centered around exosomes [20].

### 3.1. Composition of Exosomes

Exosomes are small EVs comprising a lipid bilayer membrane, transmembrane proteins, and a hydrophilic core containing proteins, mRNAs, and miRNAs. These vesicles are generated intracellularly within multivesicular endosomes, also known as MVBs. On the other hand, microvesicles (ranging from 100 to 1000 nm in diameter) are released from the surface of the plasma membrane. The multifaceted functions of exosomes encompass various biological processes, including angiogenesis, antigen presentation, apoptosis, and inflammation [23,24]. Through the transfer of informative substances, exosomes exert influence on physiological and pathological processes associated with cancer, neurodegenerative diseases, infections, and autoimmune diseases [15,25,26,27,28,29,30].

Exosomes commonly contain proteins, lipids, and nucleic acids as essential components. These constituents play a crucial role in intercellular signal transduction and the regulation of gene expression in related tissues and cells [31,32]. Notably, proteins found in exosomes include tetraspanins, heat shock proteins, lipid raft proteins, and cytoskeletal proteins, as well as proteins involved in multivesicular body formation, membrane transport and fusion, antigen presentation, and adhesion. These proteins actively participate in cell membrane fusion and exosome release [33,34]. Tetraspanins in this protein family, for example, CD9, CD63, and CD81 interact or coordinate with other proteins, contributing to membrane compartmentalization [35]. Additionally, these specific tetraspanins are abundant in EVs and are commonly employed as protein markers for characterizing EVs [36]. Furthermore, a growing body of evidence indicates the presence of various transporters and enzymes in EVs, indicating their complete functionality [15,28,37]. This suggests a potential connection between the components of EVs and the in vivo fate of drugs. The lipids in EVs are predominantly cholesterol, ceramide, phosphatidylserine, phosphatidylinositol, phosphatidylcholine, sphingomyelin, and ganglioside. These lipids play a pivotal role in the biological functions of exosomes [38,39,40]. Moreover, beyond their characteristic lipid composition, exosomes can also carry bioactive lipids such as prostaglandins, which may enhance their functional characteristics [41]. Nucleic acids present in EVs include DNA, mRNA, miRNA, long non-coding RNA (lncRNA), and circular RNA (circRNA). Nucleic acids participate in the transmission of genetic information and hold diagnostic potential [29,30,42,43,44]. Research has provided evidence of oncogene amplification and functional DNA fragments within EVs, reflecting the genetic characteristics of the parent tumor cells [45,46]. Remarkably, these transposable elements can be encapsulated and transferred from tumor cells to normal cells [47]. Furthermore, both mRNAs and miRNAs found in the exosomal fraction retain their functionality when transferred to recipient cells, highlighting the significance of exosomal RNA transfer as an important means of epigenetic signaling between cells [48,49]. Moreover, a considerable number of lncRNAs can be transported in EVs, triggering signal transmission and inducing changes in phenotypes across various cells within the TME [50,51]. Additionally, an intriguing discovery unveiled the presence of over 1000 circRNAs in EVs derived from human serum. Notably, several of these circRNAs were more abundant in EVs than in their donor cells, suggesting their potential usefulness in biomarker exploration [52,53]. The specific characteristics of exosomes are presented in Table 1.

### 3.2. Secretion of Exosomes

Exosomes are released into the extracellular space via exocytosis, a fundamental and widespread cellular process that involves the fusion of intracellular compartments with the plasma membrane [54]. In most cases, intracellular membrane fusion is mediated by specific protein mechanisms. For soluble factors, the N-ethylmaleimide-sensitive factor (NSF) plays a crucial role, while membrane complex factors rely on the participation of soluble NSF adhesion protein (SNAP) and SNAP adhesion protein receptor (SNARE) [55,56,57]. This is a highly regulated process, with the activation of SNAREs and vesicle fusion being triggered by the influx of calcium ions or calcium release from intracellular stores [58,59]. After release from the donor cell, exosomes possess the capacity to fuse with the plasma membrane or enter recipient cells through the endocytic pathway; in this pathway, the exosome components are internalized and released into the cytoplasm, consequently influencing host cells by regulating specific gene expression and signaling pathways. These interactions ultimately lead to changes in cell function or phenotype. The release of exosomes is preceded by a complex biogenesis process that involves the genesis and maturation of early endosomes into late endosomes—MVBs—where some of the endosomal membrane undergoes invagination, forming intraluminal vesicles. Crucially, the endosomal sorting complex required for transport (ESCRT) machinery plays a critical role in facilitating this process [17,60]. Additionally, members of the Rab GTPases family, including Rab27a/b, Rab11, and Rab35, act as essential coordinators for the trafficking of MVBs and the secretion of exosomes (Figure 1) [61].

### 3.3. Detection of Exosomes

The initial characterization of exosomes involved the examination of their morphology and size using electron microscopy. Subsequent analytical methods included immunoblotting, mass spectrometry, DIGE, and microarrays. In addition to these biochemical approaches, atomic force microscopy and dynamic light scattering technologies have also been employed in previous studies. The protocols for isolating exosomes differ depending on the specific biological fluid from which they originate. Notably, exosomes have densities ranging from 1.10 to 1.21 g/mL, a property often utilized for further purification using sucrose density gradients or flotation on a sucrose/deuterium oxide cushion [62]. In recent years, various technologies have been introduced for the detection, isolation, enrichment, identification, and characterization of exosomes [63]. In general, the commonly employed methods for exosome analysis include (1) western blotting, which aids in the identification of specific protein markers, (2) electron microscopy, which enables the detection of structural information, and (3) nanoparticle tracking analysis, which facilitates the quantification of exosome size and concentration. As recommended by the ISEV, it is crucial to employ two or more complementary methods to assess the results of separation techniques [14]. Moreover, several kits leveraging the composition of exosomes have been developed to enable early tumor diagnosis [15,64,65]. These kits serve as valuable tools for evaluating therapeutic efficacy and prognosis. However, a consensus on a potential gold-standard exosome detection technology is still lacking. A benchmark study revealed that currently available commercial kits may co-isolate a significant quantity of non-vesicular contaminants [66].

Recent discoveries have shed light on the substantial impact of separation techniques on subsequent RNA analyses, highlighting significant variations between different methods. In alignment with the guidelines outlined by the ISEV, the Urine Task Force of ISEV presents the current cutting-edge approaches for analyzing urinary EVs [67]. They propose that exosome purification from collected biological fluids and separation from non-EV components can be accomplished using various methods, including ultracentrifugation (UC), precipitation techniques (such as polyethylene glycol (PEG)-based methods), density gradient centrifugation (DGC), size exclusion chromatography, and immunomagnetism [14]. Despite its well-established effectiveness and suitability for most types of EVs, the differential UC method is characterized by its labor-intensive nature, time-consuming process, and limited accessibility. Furthermore, recent EV kinetic models propose that bulk EV measurements have limited sensitivity in detecting small tumors, primarily due to low signal-to-noise ratios associated with such tumors [68]. To enhance the signal derived from exosomes originating from tumors, several strategies to enrich, deplete, and filter EVs are actively being developed, holding the potential to enhance early diagnostic approaches [69,70].

## 4. The Contribution of Exosomes to RCC

### 4.1. TME

The formation of tumors is greatly influenced by the surrounding TME, as tumor cells actively engage with their microenvironments to facilitate tumorigenesis and advancement [71]. Exosomes are the primary agents responsible for cell-to-cell interactions within the TME [72]. Furthermore, tumor cells utilize EVs to transport bioactive molecules, not only targeting other tumor cells but also impacting fibroblasts, endothelial cells, immune cells, and cancer stem cells (CSCs) [73]. Therefore, EVs derived from these cells also play a role in influencing tumor progression within the TME. Mounting evidence suggests that primary tumors possess the ability to indirectly shape the TME in secondary organs by releasing various cytokines, thereby facilitating the colonization of tumor cells within the bloodstream. This specific local microenvironment preceding metastasis is referred to as the PMN [74]. The development of PMN is intricately linked to the secretion of factors by the primary tumor, including tumor-secreted factors (TSFs) and tumor-secreted exosomes (TSEs). Of these, TSEs are regarded as the primary catalysts for the formation of the PMN. In addition to tumor cells, there is a significant presence of stromal cells and immune cells in the PMN of clear cell RCC. Of these cell types, cancer-associated fibroblasts (CAFs) constitute the primary stromal components of the clear cell RCC TME [75]. Notably, it has been observed that exosomes derived from CSCs in metastatic clear cell RCC patients enhance the proliferation of clear cell RCC cells and facilitate lung metastasis [76]. Furthermore, a study demonstrated a notable disparity in the proportion of exosomes in the bloodstream of metastatic clear cell RCC patients and non-metastatic clear cell RCC patients, with the former exhibiting a significantly higher proportion [77].

Renal CSC EVs contain a diverse array of miRNAs that are potentially implicated in cellular pathways associated with cell growth and cell matrix adhesion [76]. Utilizing next-generation sequencing, a research investigation disclosed that the levels of 786-O and ACHN miR-30c-5p contained within EVs were notably lower in RCC cell lines in comparison to a human renal proximal tubular cell line, HK-2. Correspondingly, a distinctive miR-30c-5p expression pattern was evident in urinary EVs obtained from healthy individuals and clear cell RCC patients. The findings revealed that miR-30c-5p directly targeted heat-shock protein (HSP)-5, and a further study demonstrated that augmenting miR-30c-5p expression effectively hindered clear cell RCC progression both in vitro and in vivo [78].

Liu and colleagues discovered that miR-224-5p from exosomes derived from CAFs can be transferred to clear cell RCC cells and then subsequently secreted in exosomal form. miR-224-5p upregulation led to a significant increase in the number of migrating and invading clear cell RCC cells [79]. A recent report highlighted that exosomes derived from CAFs directly hinder the expression of ring finger protein 43 (RNF43) and activate the Wnt/β-catenin signaling pathway by delivering miR-181d-5p [80]. Lastly, Huang and colleagues discovered that clear cell RCC-derived exosomes containing circSAFB2 contribute to the polarization of M2 macrophages via the miR-620/JAK1/signal transducer and activator of transcription 3 (STAT3) axis, thereby remodeling the TME and promoting clear cell RCC metastasis [81].

Hypoxia in the TME of RCC is widely acknowledged. In addition to regulating the tumorigenic potential of epithelial cells, hypoxia plays a role in the generation of EVs by tumor cells in response to low pH and oxidative stress [82,83]. Wang and colleagues conducted a study demonstrating that acute hypoxic conditions induced by CoCl_2_ treatment resulted in increased miR-210 expression within EVs derived from both normal renal cells and RCC cells, with a more pronounced effect observed in the metastatic RCC cell line [84]. Overall, EVs derived from stromal and tumor cells, induced by hypoxia, play critical roles as mediators of tumorigenesis and in the reconstruction of the TME.

### 4.2. Angiogenesis

Endothelial cells are activated by tumor cells to facilitate angiogenesis. In RCC, EVs were found to contain an abundance of azurocidin protein, which plays a role in increasing vascular permeability. Consequently, tumor-derived EVs were observed to induce alterations in the phenotypes of endothelial cells, leading to disruptions in vascular morphology [85]. Moreover, it has been demonstrated that specific conditions such as hypoxia and stimulation by angiogenic growth factors enhance the release of exosomes with vasculogenic potential [86,87]. Additionally, a specific subset of tumor-initiating cells in RCC that express the mesenchymal stem cell marker CD105 can release EVs. These EVs, in turn, have the potential to induce angiogenesis and facilitate the formation of the PMN [76]. Furthermore, in clear cell RCC, exosomes derived from tumor cells contribute to angiogenesis by transporting various miRNAs and protein molecules.

A recent investigation revealed that the gene secreted frizzled-related protein 1 (SFRP1) was downregulated while miR-27a was upregulated in clear cell RCC cells [88]. Interestingly, it has been observed that miRNAs delivered by tumor-derived exosomes may also have a role in suppressing clear cell RCC angiogenesis. EVs have been linked to other tumor-promoting functions, specifically the development of resistance to TKI.

In RCC resistant to TKI, exosomal miR-549a is secreted at low levels, which in turn induces vascular permeability and angiogenesis and ultimately promotes the metastasis of RCC [89]. Additionally, Qu and colleagues demonstrated that lncRNAs associated with EVs competitively bind to miR-34 and miR-449, thereby promoting the expression of AXL and c-MET in RCC cells [90]. Consequently, they proposed EV-associated lncRNA as both a biomarker and a potential therapeutic target for combating sunitinib resistance. Li and colleagues provided evidence indicating that exosomes derived from clear cell RCC can facilitate the transfer of ApoC1 from clear cell RCC cells to tumor vascular endothelial cells. This transfer, in turn, activates the transcription factor STAT3, promoting angiogenesis and facilitating the metastasis of clear cell RCC cells [91].

Furthermore, studies have revealed that exosomes released from hypoxic clear cell RCC cells contain carbonic anhydrase 9 (CA9), which enhances angiogenesis within the TME, consequently driving cancer progression [92]. The establishment of a vascular network is crucial not only for the normal growth of tumor tissues but also provides a vital pathway for tumor invasion [93].

After discovering that renal CSC EVs carry proangiogenic miRNAs, Grange and colleagues examined the proangiogenic effects of renal CSC EVs [76]. Both renal CSC EVs and EVs obtained from non-stem RCC populations were efficiently taken up by endothelial cells. However, only renal CSC EVs had a notable impact on angiogenic processes. Specifically, renal CSC EVs stimulated the formation of capillary-like structures in Matrigel, facilitated cell invasion through Matrigel-coated transwells, and conferred resistance to apoptosis following doxorubicin treatment. Conversely, EVs derived from non-stem RCC cells did not exhibit such effects [94].

### 4.3. Immune Escape

Tumor immune escape represents a significant mechanism that drives tumor development. One factor contributing to immune escape is the ability of exosomes to diminish the cytotoxicity of natural killer (NK) cells and suppress the production of immune regulatory cytokines such as interleukin (IL)-2, interferon gamma (IFN-γ), IL-6, and IL-10. These actions ultimately facilitate immune evasion and further the progression of RCC [95]. Conversely, exosomes derived from antigen-presenting cells (APCs) carry MHC class I and class II complexes along with costimulatory proteins. When purified, these exosomes can activate CD4+ or CD8+ T cells, thus inducing anti-tumor immune responses [96].

The existing literature contains evidence of the capacity of renal CSC EVs to influence the behavior and differentiation of monocyte-derived dendritic cells. Grange and colleagues conducted a study to assess the impact of renal CSC EVs and EVs derived from non-stem RCC on dendritic cell differentiation by exposing monocytes to these EVs [97]. The presence of renal CSC EVs affected the phenotypes of monocyte-derived cells, leading to a decrease in the expression of activation markers such as CD83 and CD40, costimulatory molecules such as CD80 and CD86, the antigen-presenting molecule HLA-DR, as well as adhesion molecules involved in T-cell contact. Notably, the presence of HLA-G, which is recognized for its ability to inhibit the function of NK cells, T cells, and dendritic cells, was associated with cancer immune evasion [98]. Furthermore, approximately 50% of clear cell RCC have upregulated HLA-G, and soluble HLA-G (sHLA-G) has been detected in the plasma of patients [99]. When an antibody-blocking sHLA-G was added to monocyte-derived cells incubated with renal CSC EVs, a partial reversal of the inhibitory effect was observed. This resulted in an increase in the expression of CD86, HLA-DR, CD1a, and α5 integrin on monocyte-derived cells.

Xia and colleagues showed that EVs derived from primary RCC cells contain transforming growth factor-beta (TGF-b), a prominent immunosuppressive cytokine. Co-culturing these EVs with NK cells further intensified the dysfunction of NK cells in a TGF-b/SMAD-dependent manner [100].

RCC-derived exosomes have been shown to contain high levels of HSP-70. Similar findings were observed in exosomes secreted by mouse RCC cells (Renca cells). The presence of HSP-70 in these exosomes resulted in the upregulation of arginase 1 (ARG-1), inducible nitric oxide synthase (iNOS), IL-6, and VEGF. Additionally, HSP-70 induced the expression of myeloid-derived suppressor cells through the activation of the phosphorylated-STAT3 pathway. Ultimately, these mechanisms contributed to the promotion of tumor growth [72,101].

More recently, there has been increasing interest in the use of immune checkpoint inhibitors that target programmed death-1 (PD-1) and its ligand (PD-L1). These inhibitors have demonstrated anti-cancer effects and long-lasting improvements in conditions such as melanoma, lymphoma, bladder cancer, non-small-cell lung cancer, RCC, and other malignancies [102,103,104]. However, the overall response rate to anti-PD-1/PD-L1 therapy remains relatively low, ranging from approximately 10% to 30% [105]. To address this challenge, several innovative techniques have been developed to quantify the level of PD-L1 in EVs. These new approaches offer enhanced sensitivity, reduced time requirements, and ease of operation compared with the conventional enzyme-linked immunosorbent assay (ELISA)-based methods [106,107].

However, there is currently a lack of research focusing on PD-L1 in EVs derived from RCC. Molecules carried by EVs, including PD-L1, have the potential to serve as reliable biomarkers for immunotherapies. Tumor-derived EVs can counteract the effects of anti-PD-1 checkpoint therapy and induce systemic immunosuppression through two mechanisms [108]. The first mechanism involves direct endogenous exosomal PD-L1, where PD-L1 is present on the surface of tumor-derived EVs and is associated with tumor progression in various cancer types [109]. The second mechanism, known as indirect exosome-induced PD-L1, involves exosomes inducing the expression of PD-L1 on recipient cells, thereby indirectly regulating the immune system [110]. Notably, miR-224-5p found in urinary EVs has been shown to regulate the expression of PD-L1 in RCC cells, thereby enhancing the resistance of RCC cells to T cell-dependent toxicity [111].

### 4.4. Cancer Cell Invasion and Metastasis

Epithelial–mesenchymal transition (EMT) refers to the transformation of epithelial cells into mesenchymal cells and is characterized by the loss of polarity. EMT plays a crucial role in tumor invasion and distant metastasis [112]. This process involves the disruption of cell adhesion junctions and alterations in gene expression, ultimately leading to enhanced invasive capabilities and metastatic potential of tumor cells [113]. Numerous studies have highlighted the ability of exosomes derived from clear cell RCC cells to facilitate the EMT process in clear cell RCC. These exosomes carry diverse cargo molecules that contribute to the promotion of clear cell RCC metastasis [114].

CSCs possess the ability to differentiate into various cell types within a tumor, driving tumor development, growth, and the formation of metastases [115]. Within the context of RCC, renal CSCs exhibit self-renewal capabilities and contribute to tumor vasculogenesis, the development of metastases, and resistance to treatment [116]. In 2008, Bussolati and colleagues identified renal CSCs as a distinct cell population comprising less than 10% of the tumor mass, characterized by the presence of the mesenchymal marker, CD105 [117]. These cells also express other mesenchymal stem cell markers, including CD73, CD90, CD44, CD29, CD146, and vimentin, as well as the embryonic renal marker, Pax2. Additionally, they express embryonic stem cell markers such as OCT4, NANOG, Nestin, and Musashi, but lack differentiative epithelial markers such as cytokeratin, and adult renal progenitor markers such as CD133. The researchers proposed the hypothesis that renal CSCs could originate from mesenchymal-like resident renal stem cells or dedifferentiated embryonic progenitor cells [116]. Zhong and colleagues employed a selection process to isolate renal CSCs from an RCC cell line, focusing on their capability to grow in suspension and form spherical structures [118]. These cells exhibited positive expression of CD105, along with several stem cell-related genes such as OCT4, β-catenin, BMI, and NANOG. The research conducted by Lindoso and colleagues revealed that EVs released by CD105+ CSCs in the kidney had the ability to induce angiogenesis, both in laboratory settings and in living organisms, and also enhanced the formation of lung metastases following the intravenous injection of RCC tumor cells [119]. Wang and colleagues demonstrated that in patients with metastatic clear cell RCC, exosomes derived from CD103+ CSCs facilitated the process of EMT by transporting miR-19b-3p to cancer cells and suppressing the expression of the PTEN gene [77].

Analysis of the EMT markers N-cadherin, Vimentin, and Twist using quantitative methods indicated notable alterations in expression levels. Notably, EVs derived from CD103+ CSCs obtained from RCC patients with lung metastasis had a significant impact on EMT. Flow cytometry analysis showed that the proportion of CD103+ EVs in relation to the total EVs was higher in blood samples collected from RCC patients with lung metastasis compared with those with no metastasis [77].

In addition, tetraspanins and integrins play a significant role in determining organ-specific metastasis [120,121]. Specifically, integrins a6 and av are closely associated with lung and liver metastases, respectively [122]. Considering that the lungs and liver are frequent target sites for RCC metastasis, it is plausible that RCC-derived EVs may contain integrins a6 and av, enabling these EVs to specifically target and interact with these organs.

Recent studies have highlighted the upregulation of miR-15a in exosomes derived from clear cell RCC cells. Notably, the presence of exosomal miR-15a has been shown to enhance the activity of EMT in clear cell RCC. This effect is achieved through the downregulation of the BTG2 gene and promotion of the PI3K/AKT signaling pathway [123]. Furthermore, Hu and colleagues discovered that exosomal lncHILAR is associated with gene bank markers associated with EMT, as indicated by gene enrichment analysis. Remarkably, under both normoxic and hypoxic conditions, knocking down lncHILAR reversed the EMT process, suggesting that exosomal lncHILAR contributes to clear cell RCC metastasis by inducing EMT [124]. In their study, Li and colleagues highlighted the transfer of apolipoprotein C1 (ApoC1) from RCC cells to vascular endothelial cells through exosomes. This transfer was found to play a crucial role in promoting the metastasis of RCC cells by activating the STAT3 signaling pathway [91].

## 5. The Role of Exosomes in the Diagnosis of RCC

As research on tumor-derived exosomes in clear cell RCC advances, their potential as valuable targets in the diagnosis of clear cell RCC is becoming increasingly apparent. The utilization of radiological techniques and immunohistochemical analyses has made early detection of clear cell RCC attainable and improved prognosis [125]. However, it is worth noting that approximately 30% of clear cell RCC patients are diagnosed with metastatic disease [126,127]. Therefore, there is a critical need to incorporate specific biomarkers suitable for clear cell RCC screening and monitoring into clinical practice [128]. Liquid biopsies are generally preferred over tissue biopsies due to their less invasive nature [129]. In addition to VEGF, VEGF receptor-2 (VEGFR)-2, and CA9, which are regulated by hypoxia-inducible factor 1 subunit alpha (HIF-1a), exosomes have emerged as a novel source of non-invasive tumor biomarkers. The unique bilayer membrane structure of exosomes offers protection against external RNases and proteases, leading to enhanced stability of the enclosed mRNAs, miRNAs, and functional proteins, thus making exosomes highly sensitive markers for disease diagnosis [18]. The cargo in tumor-derived exosomes, such as the range of miRNAs, can also serve as biomarkers for clear cell RCC in the serum and urine of patients, offering valuable targets for early detection and monitoring of the disease.

### 5.1. mRNA

Exosomes have the capacity to encapsulate and deliver significant quantities of mRNA to exert their functions in recipient cells [48]. Grange and colleagues performed molecular characterization of microvesicles and identified specific mRNAs associated with tumor progression and metastasis, including VEGF, fibroblast growth factor-2 (FGF2), angiopoietin-1 (ANGPT1), ephrin-A3 (EFNA3), metalloproteinase (MMP)-2, and MMP9 [76]. Another study found that mRNA levels of glutathione transferase alpha 1 (GSTA1), CCAAT enhancer binding protein alpha (CEBPA), and pterin-4 alpha-carbinolamine dehydratase-1 (PCBD1) were lower in urinary EVs of RCC patients compared with controls. However, the mRNA levels of these three genes returned to normal one month after nephrectomy [130]. These findings suggest that mRNA levels in urinary EVs have the potential to serve as molecular markers for the diagnosis of RCC.

### 5.2. miRNA

Numerous miRNAs carried by exosomes show distinct expression patterns between patients with RCC and healthy individuals. Grange and colleagues discovered that among the CD105+ microvesicles, 24 miRNAs, including miR-200c and miR-650, were significantly upregulated, while 33 miRNAs, including miR-100 and miR-29, were downregulated [76]. Notably, miRNAs such as miR-29a, miR-650, and miR-151 were associated with tumor invasion and metastasis. Additionally, Zhang and colleagues reported that serum samples from clear cell RCC patients exhibited notably elevated levels of exosomal miR-210 and miR-1233 compared with healthy controls, and these levels significantly decreased after nephrectomy [131]. Hence, the presence of exosomal miR-210 and miR-1233 in serum could serve as a valuable indicator in the diagnosis and monitoring of clear cell RCC patients, particularly when utilizing liquid biopsies. Wang and colleagues conducted a study examining serum exosomal miR-210 and observed its upregulation in clear cell RCC, particularly in patients with advanced tumor stage, high Fuhrman grade, and metastases [84]. Additionally, they noted that clear cell RCC patients with higher miR-210 levels in exosomes had a shorter time to disease recurrence and overall survival. Furthermore, the authors concluded that exosomal miR-210 outperformed serum miR-210 as a diagnostic tool for detecting clear cell RCC and held promise as a valuable prognostic biomarker.

Another research group conducted a sequencing analysis of exosomal miRNAs derived from plasma samples, revealing an upregulation of miR-149-3p and miR-424-3p, and a significant downregulation of miR-92a-1-5p [132]. Fujii and colleagues investigated the serum levels of exosomal miR-224 and highlighted its negative prognostic value in clear cell RCC patients [133]. They further demonstrated the promising potential of exosomal miR-224 as a prognostic biomarker for detecting micro invasion or tumor metastasis following nephrectomy in clear cell RCC patients.

Furthermore, investigations have revealed that serum exosomes display significantly increased gamma-glutamyltransferase (GGT) activity in patients with advanced RCC, distant metastases, and microvascular invasion [134]. Hence, the integration of exosomal GGT alongside conventional diagnostic methods holds the potential for enhancing the diagnosis of clear cell RCC. In a recent study, researchers identified elevated levels of exosomal MYO15A in the serum of clear cell RCC patients, which correlated with poorer prognosis and suggested its potential as a diagnostic target for clear cell RCC [135].

Exosomal miR-21-5p originating from M2 macrophages has been associated with pro-metastatic effects in RCC through the activation of the PTEN/Akt pathway. A recent study demonstrated that inhibiting miR-21-5p in M2 exosomes resulted in a decrease in the metastatic potential of RCC cells [136]. Conversely, a different research group observed that metastatic spread and tumor growth in an orthotopic mouse model of clear cell RCC were reduced following the administration of mesenchymal stem cell-derived exosomes. These effects were attributed to the presence of exosomal miR-182 derived from mesenchymal stem cells, which appeared to enhance the T-cell modulated immune response and subsequently decrease the expression of VEGF-A, thereby impeding overall tumor progression [137].

### 5.3. lncRNAs and circRNAs

LncRNAs are RNA molecules consisting of more than 200 nucleotides that regulate cellular processes such as transcription and protein translation by interacting with proteins, mRNAs, or miRNAs [138]. They exhibit specific expression patterns in tumor cells, suggesting their potential utility in cancer diagnosis [139]. Exosomes are enriched with lncRNAs, which play significant roles in the growth, proliferation, invasion, and metastasis of cancer cells [140]. Research has indicated that exosome-mediated transfer of lncARSR promotes the expression of AXL and c-MET in RCC cells by competitively binding to miR-34/miR-449, thereby contributing to the development of resistance to sunitinib [90].

circRNAs are a recently identified class of non-coding RNAs that have been detected in exosomes and act as miRNA sponges during gene regulation [52]. Notably, the analysis of the circRNA expression array revealed a significant upregulation of circ_400068 in RCC-derived exosomes [141]. However, the exploration of circRNAs in exosomes derived from renal cancer cells is an area that requires further investigation. Table 2 provides a compilation of potential biomarkers derived from exosomes that have been validated in RCC.

## 6. Exosomes in the Treatment of RCC

### 6.1. Tumour Drug Resistance

Due to the inactivation of the Von Hippel-Lindau (VHL) gene in clear cell RCC, several receptor tyrosine kinases crucial for angiogenesis and TME homeostasis are often overexpressed. Consequently, tyrosine kinase inhibitors have been approved as a first-line treatment for RCC [148,149]. However, the development of drug resistance poses a significant challenge for patients with advanced RCC [150]. Kidneys play a vital role in drug elimination and reabsorption, housing various drug transporters in the proximal tubules. Therefore, the variability in renal drug transporters can greatly impact drug disposition processes [151]. Drug resistance can be horizontally transmitted between cells through the transfer of EVs carrying various cargo such as drug-efflux transporters, miRNAs, and lncRNAs [152]. The variability in renal drug transporters plays a crucial role in influencing drug disposition processes [151].

Sunitinib, an oral multi-targeted TKI known for its potent antiangiogenic properties, shows limited effectiveness in approximately 25% of clear cell RCC patients, and most patients experience a relapse within one year of treatment [153]. A recent study discovered that lncARSR plays a role in inducing resistance to sunitinib in initially sensitive RCC cells [90]. The researchers observed that drug-resistant cells in nephropathy transferred lncARSR to neighboring cells through exosomes, leading to the development of drug resistance in those cells. By competitively binding to miR-34/miR-449, lncARSR enhances the expression of AXL and c-MET and activates the STAT3, AKT, and ERK signaling pathways. The activation of AKT further amplifies lncARSR expression by inhibiting the transcription factors FOXO1 and FOXO3a, establishing a positive feedback loop. Additionally, Rab27b, a key protein involved in exosome secretion, may contribute to the development of drug resistance through the MAPK and VEGF signaling pathways [154]. In this context, Rab27b may play an oncogenic role in sunitinib resistance in RCC, independent of its involvement in exosome-mediated mechanisms.

Sorafenib is an orally administered multikinase inhibitor that effectively targets growth signaling and angiogenesis in clear cell RCC by blocking several key receptors, including VEGFR-2, VEGFR-3, platelet-derived growth factor (PDGF)-beta receptor (PDGFR-b), RAF-1, c-Kit protein (c-Kit), and FMS-like tyrosine kinase 3 (Flt-3) [155]. Research has shown that exosomes derived from tumor cells can transfer drug resistance information from sorafenib-resistant clear cell RCC cells to non-resistant clear cell RCC cells by delivering miR-31-5p, which targets the 3’-UTR region of the human mutL homolog 1 (MLH1) gene [156]. Interestingly, Xuan and colleagues discovered that miR-549a expression was lower in TKI-resistant clear cell RCC cells and their associated exosomes compared with TKI-sensitive clear cell RCC cells. Consequently, delivering miR-549a to TKI-resistant renal cancer cells could potentially reverse their resistance to TKIs [89].

Several substances, including ketoconazole and tipifarnib, have been investigated for their potential inhibitory effects on exosome biogenesis and secretion in drug-resistant cancers. These substances have undergone testing in RCC cell lines and metastatic prostate cancer cell lines [157,158,159]. Ketoconazole, for instance, has shown an ability to reduce tumor-specific exosomes by inhibiting the expression of Alix, nSMase, and Rab27a proteins (Figure 2). This inhibition leads to a decrease in the delivery of substances carried by exosomes, resulting in improved sunitinib efficacy and reduced drug resistance [159]. Understanding the mechanisms underlying the drug resistance mediated by EVs is of great importance as it may contribute to the identification of novel prognostic and predictive biomarkers.

### 6.2. Tumour Vaccines

Numerous studies have examined the function and mechanism underlying exosome-based immunotherapy in oncology and its significant therapeutic effect on cancer progression, with a prime focus on the potential development of immunotherapeutic vaccines [161]. In many investigations, exosomes derived from dendritic cells have been utilized as carriers for anti-tumor agents like peptides or as effectors following the stimulation of dendritic cells with specific cancer biomarkers [162].

A report by Zhang and colleagues showed that exosomes derived from RCC cells, which were anchored with IL-12, stimulated the production of cytotoxic T lymphocytes targeting RCC antigens and demonstrated improved anti-tumor effects [163]. The researchers developed the EXO-IL-12 vaccine capable of expressing the kidney cancer-specific antigen G250, along with immune-associated protein and GPIIL-12, which significantly enhanced the proliferation and activation of T lymphocytes in vitro [164]. Another study observed that mice with RCC vaccinated with a dendritic cell vaccine loaded with tumor exosomes (DC-TEX) had a longer survival period compared with mice vaccinated with a dendritic cell vaccine loaded with tumor cell lysates [165].

## 7. Conclusions and Future Directions

Early diagnosis plays a pivotal role in improving the survival rate of RCC patients, and exosomes offer potential benefits in this regard. Due to their small size, high mobility, and lipid bilayer structure, exosomes can readily traverse biological membranes and safeguard the bioactive cargo enclosed within their membranes from degradation. Exosomes play a significant role in the invasion and metastasis of RCC and contribute to tumor drug resistance and immune evasion.

This review presents the state-of-the-art biological roles and translational relevance of tumor-derived exosomes in various aspects of RCC cancer, including PMN formation, tumor angiogenesis, and EMT during the progression of metastasis. Additionally, we explored the role of tumor-derived exosomes in mediating drug resistance in RCC patients through the delivery of miRNA, lncRNA, and protein molecules. The cargo transported by exosomes, particularly the range of miRNAs, holds potential as diagnostic biomarkers for RCC, offering valuable targets for early detection and monitoring of the disease, as well as potential interventions.

Currently, research on exosomes is primarily confined to preclinical investigations and early-stage clinical trials, with challenges in translating experimental findings into clinical applications. To facilitate the utilization of exosomes in clinical settings, further comprehensive studies integrated with clinical trials are warranted. Future investigations should encompass larger sample sizes and diverse tissue types and employ prospective study designs that yield more compelling evidence and robust medical data to support clinical translation. Furthermore, the examination of exosomes in the context of RCC has been relatively isolated, and none of the identified molecules have undergone consistent validation across multiple studies. This necessitates the need for additional prospective clinical trials to establish more reproducible biomarkers. An increasing number of studies are expected to concentrate on exploring the utilization of tumor-derived exosomes in liquid biopsy and treatment strategies for RCC. Finally, further investigations are warranted to develop exosome-mediated tumor vaccines and gain a deeper understanding of the impact of exosomes and mechanisms of drug resistance in targeted therapies for RCC. In conclusion, while much progress has been made, further research is necessary to comprehend the precise role of tumor-derived exosomes in RCC diagnosis, drug resistance, and metastasis. Only through multidisciplinary efforts will tumor-derived exosomes in RCC be transformed into a tangible reality and integral part of diagnostic and personalized medicine. The utilization of AI technologies for omics analysis is expected to progressively increase in the future, given the realization of precision medicine as a goal all over the world.

## Figures and Tables

**Figure 1 ijms-24-14356-f001:**
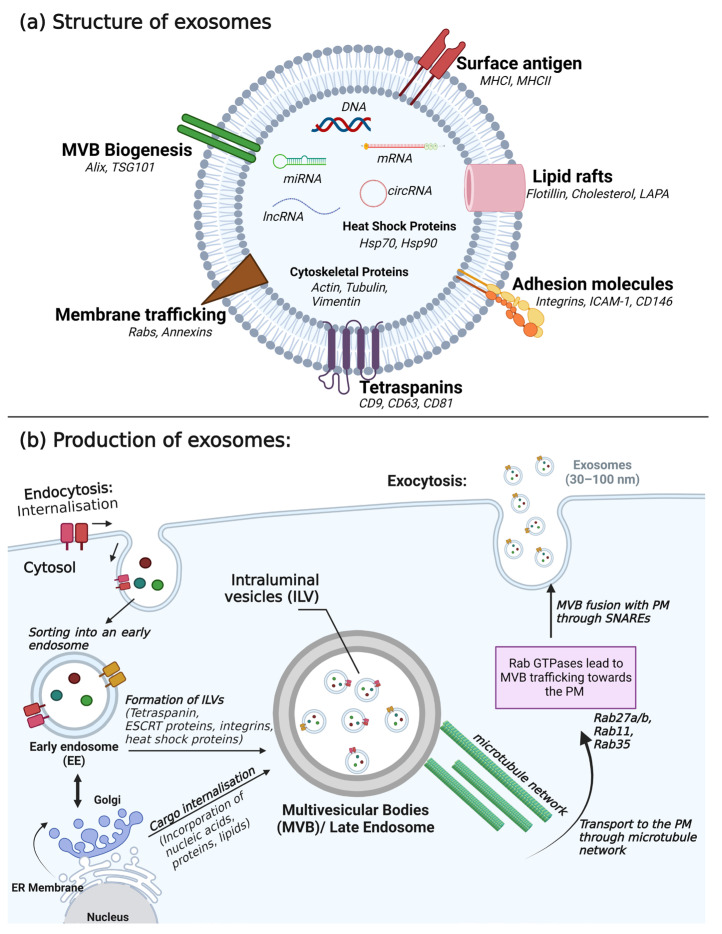
Structure and production of exosomes. (**a**) Structure of exosome depicting the molecular components; the phospholipid bilayer (blue colored) encloses the vesicle. Internally, the vesicle consists of cargo comprising nucleic acids, proteins, and lipids. (**b**) The biogenesis of exosomes involves (1) cargo from the extracellular space entering the cytosol through internalization wherein it forms the early endosome. (2) Early endosomes mature to form multivesicular bodies (MVBs) consisting of intraluminal vesicles that include tetraspanin (CD9, CD63, and CD81), an endosomal sorting complex required for transport (ESCRT) proteins (Alix, TSG101), integrins, heat shock proteins, cytoskeletal proteins, and membrane transport proteins. (3) Cargo internalization consists of cargo delivery from the trans-Golgi network and the cytosol. (4) MVBs consisting of exosome cargo are transported to the plasma membrane (PM) with the help of a microtubule network and Rab GTPases. (5) SNAREs dock the MVBs to the PM and ILVs are secreted as exosomes [21,22]. Created with BioRender.com, accessed on 15 August 2023.

**Figure 2 ijms-24-14356-f002:**
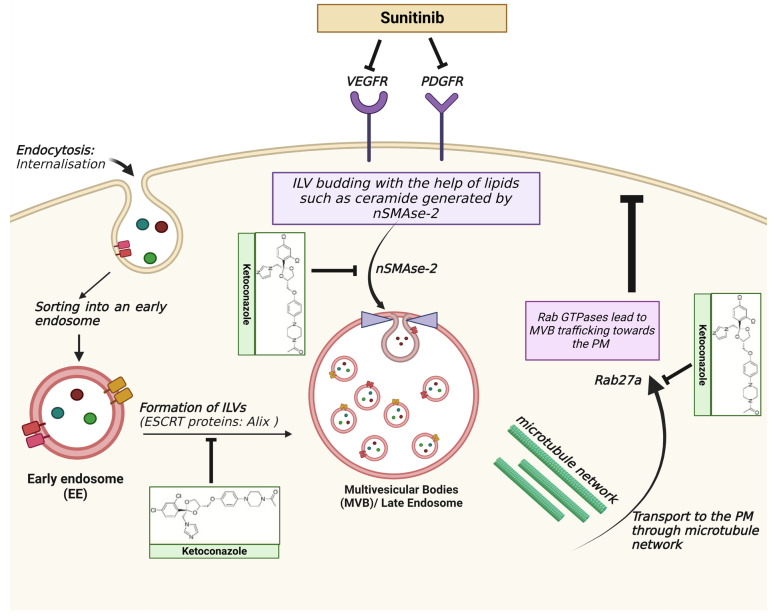
The RTK inhibitor sunitinib in combination therapy with ketoconazole in RCC. The exosomes secreted from tumor cells affect the cells in the tumor microenvironment, leading to drug resistance and immune escape. The ESCRT pathway is one of the central components responsible for the biogenesis of exosomes. Lipids such as ceramide are generated by neutral sphingomyelinase (nSMAse-2), which is necessary for vesicle budding during ILV formation. Rab GTPases lead to the docking of MVBs to the plasma membrane (PM). Ketoconazole leads to the downregulation of Alix, nSMAse-2, and Rab27a, inhibiting exosome biogenesis and secretion and leading to a decrease in tumor growth [159,160]. Created with BioRender.com, accessed on 15 August 2023. Abbreviations—VEGFR: vascular endothelial growth factor receptor; PDGFR: platelet-derived growth factor receptor: ILV: intraluminal vesicles: ESCRT: endosomal sorting complex required for transport; MVB: multivesicular bodies: PM: plasma membrane.

**Table 1 ijms-24-14356-t001:** Characteristics of exosomes.

Size	30–100 nm	
Formation and release	Formed intracellularly within multivesicular bodies	
Isolation and detection	Ultracentrifugation, electron microscopy, western blotting, mass spectrometry, and nanoparticle tracking analysis	
Proteins	Type	Cargo
	Tetraspanin	CD9, CD63, CD81
	Heat shock protein	Hsp70, Hsp90
	Membrane transport and fusion proteins	Rab, Annexins (I, II, IV, V, VI)
	Antigen presentation	MCH I, MCH II
	Adhesion molecules	Integrins, ICAM-1, CD146
	Lipid raft	LAPA, Flotillin-1, Cholesterol
	Cytoskeletal proteins	
Lipids	Cholesterol, ceramide, phosphatidylserine, phosphatidylinositol, phosphatidylcholine, sphingomyelin, ganglioside	
Nucleic acids	DNA, mRNA, miRNA, lncRNA, circRNA	

**Table 2 ijms-24-14356-t002:** Potential EV-derived biomarkers with clinical significance in RCC.

Type	EV Source	EV Cargoes	Analysis Method	Cohorts	Clinical Significance	Ref.
mRNA	Supernatants	VEGF, FGF2, ANGPT1, EFNA3, MMP2, and MMP9	Microarray, qRT-PCR	Cancer stem cells	mRNAs implicated in tumor progression and metastasis through molecular characterization of VEGF, FGF2, ANGPT1, EFNA3, MMP2, and MMP9.	[76]
	Urine	*GSTA1*, *CEBPA*, and *PCBD1* genes	Microarray, qRT-PCR	46 RCC patients (33 with ccRCC), 22 HS	Significantly lower in ccRCC patients compared with HS. Increased to normal level 1 month after nephrectomy.	[130]
miRNA	Supernatants	miR-200c, miR-650	Microarray, qRT-PCR	Cancer stem cells	miR-200c and miR-650 significantly upregulated in CD105+ microvesicles.	[76]
		miR-100, miR-296			miR-100 and miR-296 significantly downregulated.	
		miR-29a, miR-650, miR-151			miR-29a, miR-650, and miR-151 are associated with tumor invasion and metastasis.	
		miR-549a	qRT-PCR	786-O cell, 786-O-SR cell	Lower in TKI-resistant cells and exosomes.	[89]
		miR-205	qRT-PCR	HK-2 cell, 786-O cell	Significant differences in concentration between the two cell lines.	[142]
	Urine/Supernatants	miR-204-5p	qRT-PCR	Mouse and human tRCC cell lines	Significantly increased in primary RCC cell lines established from transgenic mouse tumors and tumor tissue from 2 Xp11 tRCC patients.	[143]
	Urine	miR-126-3p	Microarray, qRT-PCR	81 ccRCC patients, 33 HS	Differentiated ccRCC patients from HS	[144]
		miR-126-3p combined miR-449a				
		miR-126-3p combined miR-34b-5p			Differentiated patients with ccRCC and small renal masses (pT1a, ≤4 cm) from HS.	
		miR-126-3p combined miR-486-5p		24 benign renal tumor patients, 33 HS	Differentiated benign patients from HS.	
		miR-30c-5p	RNA sequencing, qRT-PCR	70 early-stage ccRCC patients, 30 HS	Significantly lower in early-stage ccRCC patients compared with HS.	[78]
	Plasma	miR-92a-1-5p, miR-149-3p, miR-424-3p	RNA sequencing, qRT-PCR	5 RCC, 5 healthy controls. 22 RCC, 16 healthy controls	Significantly downregulated in the plasma exosomes of RCC.	[132]
		miR-15a	Microarray, qRT-PCR	ACHN cell	Upregulated in ccRCC cells. *BTG2* gene is negatively correlated with miR-15a expression.	[123]
		miR-let-7i-5p, miR-26a-1-3p, miR-615-3p	RNA-sequencing, qRT-PCR	A cohort of 44 metastatic RCC patients for screening and 65 validation controls	Low levels correlated with poor OS of metastatic RCC patients.	[145]
	Serum	miR-1233, miR-210	qRT-PCR	82 ccRCC patients, 80 HS	Significantly higher in ccRCC patients than in HS	[131]
		miR-210	Microarray, qRT-PCR	45 pre-operative and 35 post-operative ccRCC patients, 30 HS	Significantly higher in ccRCC patients compared with HS, Significantly higher in pre-operative than post-operative samples.	[84]
		miR-224	qRT-PCR	108 ccRCC patients	High levels are correlated with shorter PFS, CSS, and OS of ccRCC patients	[133]
Protein	Urine	MMP9, CP, PODXL, CAIX, and DKK4	LC-MS/MS, western blotting	29 RCC, 23 healthy controls	MMP9, CP, PODXL, CAIX, and DKK4 are higher in patients with RCC compared with HS.	[146]
		CD10, EMMPRIN, DPEP1, syntenin 1, and AQP1			CD10, EMMPRIN, DPEP1, syntenin 1, and AQP1 are higher in HS than in patients with RCC	
	Serum	CD103	Flow cytometry	76 metastatic and 133 non-metastatic ccRCC patients	Higher ratio of CD103^+^ EVs to total EVs in samples from metastatic patients compared with samples from non-metastatic patients.	[77]
		Azurocidin	LC-MS/MS	19 ccRCC patients, 10 HS	Significantly higher in ccRCC patients than in HS	[85]
	Tissue	Azurocidin	LC-MS/MS	20 paired tumors and adjacent normal tissues from ccRCC patients	Significantly higher in ccRCC patients than in HS	[85]
	Supernatants	RAB27B	ExoELISA-ULTRA CD63 kit	A498 cell	Oncogenic role in RCC and sunitinib resistance.	[110]
IncRNA	Plasma	Circulating lncARSR	qRT-PCR	71 advanced ccRCC patients, 32 HS	Poor sunitinib response, Sunitinib resistance via competitively binding miR-34/miR-449.	[90]
circRNA	Plasma	circ_400068	CircRNA microarray	28 RCC	Upregulated in RCC plasma exosomes, tissue samples, and cells.	[141]
Lipid	Urine	LysoPE	microLC-Q-TOF-MS	8 ccRCC patients, 8 HS	48 differential lipidome expression (22 upregulated and 26 downregulated) in ccRCC.	[147]

Abbreviations—EV: extracellular vesicles; RCC: renal cell cancer; ccRCC: clear cell renal cell cancer; qRT-PCR: quantitative reverse transcription polymerase chain reaction; PFS: progression-free survival; VEGF: vascular endothelial growth factor; FGF2: fibroblast growth factor 2; ANGPT1: angiopoietin-1; EFNA3: ephrin A3; MMP2: matrix metallopeptidase 2; MMP9: matrix metallopeptidase 9; Xp11 tRCC: Xp11.2 translocation renal cell carcinoma; TKI: tyrosine kinase inhibitors; OS: overall survival; CSS: cancer-specific survival; CP: ceruloplasmin; PODXL: podocalyxin like; CAIX: carbonic anhydrase IX; DKK4: dickkopf 4; EMMPRIN: extracellular matrix metalloproteinase inducer; DPEP1: dipeptidase 1; AQP1: aquaporin 1; LC-MS/MS: liquid chromatography-tandem mass spectrometry.

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
