# Peer review of "Exosomes in the Diagnosis and Treatment of Renal Cell Cancer"

_ijms, 2023, doi:10.3390/ijms241814356_

Round 1

Reviewer 1 Report

I have had the opportunity to thoroughly review the manuscript titled "Exosomes in the Diagnosis and Treatment of Renal Cell Cancer," authored by Stergios Boussios. I commend the authors for their comprehensive and well-written contribution to the understanding of the role of exosomes in the context of renal cell cancer (RCC). I believe that this manuscript holds substantial value for both researchers and clinicians in the field of oncology and exosome biology. I recommend its publication after addressing a few minor revisions.

Strengths:

Comprehensive Literature Review: The manuscript provides an in-depth and up-to-date review of the existing literature on exosomes and their implications in the diagnosis and treatment of RCC. This review is well-structured and logically organized, allowing readers to grasp the key concepts and findings easily.

Clarity of Writing: The authors have displayed commendable writing skills, presenting complex scientific information in a clear and understandable manner. This will be especially helpful for readers who might not be well-versed in the field of exosome research.

Critical Analysis: The manuscript does an excellent job of critically analyzing the studies and research findings it reviews. The authors have not only summarized the current state of knowledge but have also offered insightful discussions on controversies, limitations, and potential future directions.

Clinical Relevance: The manuscript effectively highlights the clinical relevance of exosome research in RCC, discussing potential applications in early diagnosis, prognosis, and treatment strategies. This aspect will likely capture the attention of both researchers and medical practitioners.

Figure and Table Utilization: The incorporation of figures and tables significantly enhances the readability and impact of the manuscript. The visual representations effectively complement the text, aiding in the understanding of complex concepts.

Suggestions for Improvement:

1. Figure 1: The upper part of Figure 1, illustrating the "Structure of Exosomes," would greatly benefit from additional detail and elaboration. A higher resolution image should be considered to ensure clarity. Additionally, it would be valuable to highlight any signature proteins or unique characteristics specific to extracellular vesicles originating from renal cells. This will enhance the figure's informational value.

2. Redundancy Deletion:

Lines 66-67: Redundant sentence; consider deleting.

Lines 87-89: Redundant sentence; consider removal.

3. Conciseness Enhancement:

Lines 136-140: To enhance conciseness and clarity, consider revising the sentences to avoid redundancy.

Please check other sections too for redundancy.

4. In-text Citations Consistency and Support:

Example Line 582-586: To bolster the credibility of certain statements, it is advisable to consistently provide specific in-text citations. Ensuring a concise and focused use of references will strengthen the manuscript's scientific foundation.

Thank You.

Author Response

Dear Editor and Reviewers,

I am pleased to resubmit for publication the revised version of ijms-2589043 manuscript, entitled “Exosomes in the Diagnosis and Treatment of Renal Cell Cancer”.

Thankfully the reviewers provided us with a great deal of guidance, regarding how to better position the article. We are hopeful you agree that this revision will update our comprehensive review. All the comments have been addressed, as shown in the revised version of the manuscript, along with this point-by-point response to the reviewers' comments.

All corresponding are blue changes in the manuscript.

Reviewer #1:

  • General comment:

I have had the opportunity to thoroughly review the manuscript titled "Exosomes in the Diagnosis and Treatment of Renal Cell Cancer," authored by Stergios Boussios. I commend the authors for their comprehensive and well-written contribution to the understanding of the role of exosomes in the context of renal cell cancer (RCC). I believe that this manuscript holds substantial value for both researchers and clinicians in the field of oncology and exosome biology. I recommend its publication after addressing a few minor revisions.

Strengths:

Comprehensive Literature Review: The manuscript provides an in-depth and up-to-date review of the existing literature on exosomes and their implications in the diagnosis and treatment of RCC. This review is well-structured and logically organized, allowing readers to grasp the key concepts and findings easily.

Clarity of Writing: The authors have displayed commendable writing skills, presenting complex scientific information in a clear and understandable manner. This will be especially helpful for readers who might not be well-versed in the field of exosome research.

Critical Analysis: The manuscript does an excellent job of critically analyzing the studies and research findings it reviews. The authors have not only summarized the current state of knowledge but have also offered insightful discussions on controversies, limitations, and potential future directions.

Clinical Relevance: The manuscript effectively highlights the clinical relevance of exosome research in RCC, discussing potential applications in early diagnosis, prognosis, and treatment strategies. This aspect will likely capture the attention of both researchers and medical practitioners.

Figure and Table Utilization: The incorporation of figures and tables significantly enhances the readability and impact of the manuscript. The visual representations effectively complement the text, aiding in the understanding of complex concepts.”.

Response:

Thank you for your positive reinforcement and constructive feedback. We appreciate the opportunity to revise our work for consideration for publication.

  • Suggestions for Improvement:

  1. Figure 1: The upper part of Figure 1, illustrating the "Structure of Exosomes," would greatly benefit from additional detail and elaboration. A higher resolution image should be considered to ensure clarity. Additionally, it would be valuable to highlight any signature proteins or unique characteristics specific to extracellular vesicles originating from renal cells. This will enhance the figure's informational value.

Response:

Thank you for your comment.

We have added an elaborated figure for structure of exosomes with increased resolution. RCC-specific proteins have not been added in the figure, since there are numerous proteins excreted by the vesicles and it will be difficult for the reader to understand the image. All of the RCC-specific proteins are detailed in table 2 though. References 21 and 22 have also be incorporated.

  1. Redundancy Deletion:

Lines 66-67: Redundant sentence; consider deleting.

Lines 87-89: Redundant sentence; consider removal.

Response:

Thank you for your recommendation, which makes absolutely.

We have now deleted both sentences.

  1. Conciseness Enhancement:

Lines 136-140: To enhance conciseness and clarity, consider revising the sentences to avoid redundancy.

Please check other sections too for redundancy.

Response:

Thank you for your concern. We have now revised these sentences to avoid redundancy.

We have also checked throughout the manuscript for redundancy.

  1. In-text Citations Consistency and Support:

Example Line 582-586: To bolster the credibility of certain statements, it is advisable to consistently provide specific in-text citations. Ensuring a concise and focused use of references will strengthen the manuscript's scientific foundation.

Response:

Thank you for your recommendation. We have now added two specific in-text references (161 and 162) in this paragraph.

We have also checked throughout the manuscript for redundancy.

Reviewer 2 Report

Comments to the Author

  1. The paper effectively highlights the relevance of exosomes in the context of RCC, addressing the challenges faced in diagnosis and treatment. It underscores the potential advantages of exosomal biomarkers and exosome-based therapeutic approaches. It will be a good addition to the field and will be very helpful for researchers in expanding their knowledge and understanding. The authors provide sufficient background information and include relevant references to support the discussion.
  2. Some paragraphs are too long and can be divided into smaller paragraphs, for example: Page 7 (line 276-305), Page 8 (line 316-347), Page 9 (line 318-413) and Page 10&11 (line 459-489).
  3. Provide acronym expansion when referring to the acronym first time.
  4. A few lines on the methodology, selection criteria and search strategy for the review will be very helpful for the reader. Include references: Extracellular vesicles in renal cell carcinoma: challenges and opportunities coexist. Frontiers in Immunology, 14, 2023 & Exosomes and cancer - Diagnostic and prognostic biomarkers and therapeutic vehicle. Oncogenesis 11, 54 (2022).

5.       Page 2, line 63-71, provide reference.

  1. Page 2, line 84, change to healthy.
  2. Doublecheck reference 15, 19, 25, 26, 27, 54, 59, 93, 109, 111.
  3. Page 3, Figure 1, Structure of exosomes inset is hazy, correct for easier visualization.
  4. Page 3, Figure 1 (Structure and production of exosomes) and lines 104-112, give references.
  5. Page 4, Table 1. Characteristics of exosomes, keep table in one page, split between page 4 and 5.
  6. Page 5, line 186, correct to DIGE.
  7. Page 7, line 269, correct COCl2.
  8. Page 7, line 295, correct to c-MET.
  9. Page 11, line 364, correct clear.
  10. Page 11, line 490-493, give reference.
  11. Page 14, in abbreviation list PFS expansion missing.
  12. Page 17, line 604, check grammar.

Author Response

Dear Editor and Reviewers,

I am pleased to resubmit for publication the revised version of ijms-2589043 manuscript, entitled “Exosomes in the Diagnosis and Treatment of Renal Cell Cancer”.

Thankfully the reviewers provided us with a great deal of guidance, regarding how to better position the article. We are hopeful you agree that this revision will update our comprehensive review. All the comments have been addressed, as shown in the revised version of the manuscript, along with this point-by-point response to the reviewers' comments.

All corresponding are blue changes in the manuscript.

Reviewer #2:

Comments to the Author

  1. The paper effectively highlights the relevance of exosomes in the context of RCC, addressing the challenges faced in diagnosis and treatment. It underscores the potential advantages of exosomal biomarkers and exosome-based therapeutic approaches. It will be a good addition to the field and will be very helpful for researchers in expanding their knowledge and understanding. The authors provide sufficient background information and include relevant references to support the discussion.

Response:

Thank you for your positive feedback. We felt very appreciated after hearing your kind words about our manuscript.

  1. Some paragraphs are too long and can be divided into smaller paragraphs, for example: Page 7 (line 276-305), Page 8 (line 316-347), Page 9 (line 318-413) and Page 10&11 (line 459-489).

Response:

Thank you for your valuable comment. The sections “3.2 Angiogenesis”, “3.3 Immune escape”, “3.4 Cancer cell invasion and metastasis” and “4.2 miRNA” have been subdivided in smaller paragraphs as you kindly recommended.

  1. Provide acronym expansion when referring to the acronym first time.

Response:

Thank you for your comment. We have provided acronym expansion when referring to the acronym first time.

  1. A few lines on the methodology, selection criteria and search strategy for the review will be very helpful for the reader. Include references: Extracellular vesicles in renal cell carcinoma: challenges and opportunities coexist. Frontiers in Immunology, 14, 2023 & Exosomes and cancer - Diagnostic and prognostic biomarkers and therapeutic vehicle. Oncogenesis 11, 54 (2022).

Response:

Thank you for your valuable comment. We have now incorporated the "Methods" section, along with the recommended references (13, 162, and 128, respectively).

  1. Page 2, line 63-71, provide reference.

Response:

Thank you for your valuable comment. We have now included the reference 13.

  1. Page 2, line 84, change to healthy.

Response:

Thank you for your comment. We have now changed to “healthy”.

  1. Doublecheck reference 15, 19, 25, 26, 27, 54, 59, 93, 109, 111.

Response:

Thank you; all the references have been double checked.

  1. Page 3, Figure 1, Structure of exosomes inset is hazy, correct for easier visualization.

Response:

Thank you for your recommendation. We have recreated Figure 1 and corrected the structure of exosomes in the revised version of the manuscript. We also added references 21 and 22.

  1. Page 3, Figure 1 (Structure and production of exosomes) and lines 104-112, give references.

Response:

Thank you for your recommendation. We have now included the references 21 and 22.

  1. Page 4, Table 1. Characteristics of exosomes, keep table in one page, split between page 4 and 5.

Response:

Thank you for your recommendation. Table 1 is now in one page.

  1. Page 5, line 186, correct to DIGE.

Response:

Thank you; we have corrected to DIGE.

  1. Page 7, line 269, correct COCl2.

Response:

Thank you; we have corrected to COCl2.

  1. Page 7, line 295, correct to c-MET.

Response:

Thank you; we have corrected to c-MET.

  1. Page 11, line 364, correct clear.

Response:

Thank you; we have corrected to clear.

  1. Page 11, line 490-493, give reference.

Response:

Thank you; we have given the reference.

  1. Page 14, in abbreviation list PFS expansion missing.

Response:

Thank you; we have incorporated the PFS expansion in the abbreviation list.

  1. Page 17, line 604, check grammar.

Response:

Thank you; we have incorporated the PFS expansion in the abbreviation list.